# In Vivo Monitoring of Glycerolipid Metabolism in Animal Nutrition Biomodel-Fed Smart-Farm Eggs

**DOI:** 10.3390/foods13050722

**Published:** 2024-02-27

**Authors:** Victor A Zammit, Sang O Park

**Affiliations:** 1Metabolic Biochemistry, Warwick Medical School, University of Warwick, Coventry CV4 7AL, UK; vic.zamit01@gmail.com; 2Institute of Animal Life Science, Kangwon National University, Chuncheon-si 24341, Gangwon State, Republic of Korea

**Keywords:** egg, esterification, glycerolipid, in vivo monitoring, oxidation, phospholipid, triacylglycerol

## Abstract

Although many studies have examined the biochemical metabolic pathways by which an egg (egg yolk) lowers blood lipid levels, data on the molecular biological mechanisms that regulate and induce the partitioning of hepatic glycerolipids are missing. The aim of this study was to investigate in vivo monitoring in four study groups using an animal nutrition biomodel fitted with a jugular-vein cannula after egg yolk intake: CON (control group, oral administration of 1.0 g of saline), T1 (oral administration of 1.0 g of pork belly fat), T2 (oral administration of 1.0 g of smart-farm egg yolk), and T3 (oral administration of T1 and T2 alternately every week). The eggs induced significant and reciprocal changes in incorporating ^14^C lipids into the total glycerolipids and releasing ^14^CO_2_, thereby regulating esterification and accelerating oxidation in vivo. The eggs increased phospholipid secretion from the liver into the blood and decreased triacylglycerol secretion by regulating the multiple cleavage of fatty acyl-CoA moieties’ fluxes. In conclusion, the results of the current study reveal the novel fact that eggs can lower blood lipids by lowering triacylglycerol secretion in the biochemical metabolic pathway of hepatic glycerolipid partitioning while simultaneously increasing phospholipid secretion and ^14^CO_2_ emission.

## 1. Introduction

The biochemical metabolic partitioning of hepatic glycerolipids by eggs (phospholipid in the egg yolk) is of great interest in relation to the reduction of unhealthy blood lipids. Numerous studies have been conducted to assess the biochemical metabolic pathways through which an egg lowers blood lipids [1,2,3], but little work has been performed to elucidate the molecular biological mechanisms that regulate and induce the partitioning of hepatic glycerolipids. A new study on egg cholesterol was undertaken, as it was confirmed that saturated fatty acid (SFA) rather than dietary cholesterol increases unhealthy lipids in the blood and causes various metabolic diseases, including hyperlipidemia. Samgyeopsal (pork belly cooked on a grill), which is the most popular South Korean food, has been known to increase blood lipids due to its high content of SFA and cholesterol [4]. Nutritional strategies that promote egg intake were highlighted to dispel misconceptions about cholesterol from eggs and increase egg consumption. Although eggs are the nutritionally most complete natural food, the predominant opinion was that egg intake should be limited because of the high cholesterol content. Since eggs are low in SFAs and high in phospholipids that lower cholesterol and promote brain activity, a new study was conducted based on the hypothesis that egg intake could help reduce unhealthy blood lipids and improve the development and mental learning abilities of growing adolescents while simultaneously preventing Alzheimer’s disease [5,6].

Phospholipids (PLs; lecithin, phosphatidyl choline) present in eggs are known to lower blood lipids. Egg yolks contain 31–36% of lipids, of which one-third of the lipids are PLs and 66–80% of the PLs are in the form of lecithin [7,8]. The PL lecithin present in eggs has been shown to rapidly reduce cholesterol absorption by maintaining phospholipid saturation in the small intestine, inhibit HMG-CoA reductase activity for cholesterol biosynthesis, and promote cholesterol excretion through feces [1,5,8]. Egg PLs lower cholesterol uptake in vivo by inhibiting the mobilization of cholesterol from mixed micelles through the lymph ducts [5,8]. Eggs have been described to lower blood lipids in a number of references to specific findings or major research articles [see above]. Recently, as smart-farm technology spreads, consumers’ interest in the nutrition of smart-farm eggs (eggs produced by a digital livestock system) is increasing, and studies have also reported on a biochemical metabolic pathway that lowers blood lipids in an animal nutritional biomodel [9]. However, the molecular mechanism for hepatic glycerolipid partitioning on reducing blood lipids in an animal biomodel fitted with a jugular-vein cannula after being fed smart-farm eggs is unknown.

On the other hand, after dietary intake, lipoprotein particles and their residues can elevate triacylglycerol (TAG) in the blood via the absorption of chylomicrons that are competitive against lipoprotein lipase activity in the digestive tract, so that they can continue to secrete very-low-density lipoprotein-triacylglycerol (VLDL-TAG). Hepatic glycerolipid partitioning related to PL secretion from the liver into the blood is important to reduce blood lipids [10,11]. However, little is known about the clear molecular mechanisms of the biochemical metabolic pathway for hepatic glycerolipid partitioning associated with the reduction of blood lipids through egg intake. Therefore, it is necessary to identify the biochemical mechanism of the metabolic partitioning of hepatic glycerolipids (esterification and oxidation) related to TAG and PLs, which are newly synthesized in the liver and secreted into the blood after egg intake. Advanced molecular biology techniques, such as in vivo monitoring, are required to clearly identify the biochemical metabolic pathway of the partitioning of hepatic glycerolipid after egg intake.

In vivo monitoring is an advanced technique that is performed in the liver of awake animals fitted with a jugular-vein cannula to address the in vitro conundrum (hormonal action and trace lipids cannot be replaced in vivo) in the study to investigate the metabolic pathways of hepatic glycerolipid. The results of in vivo monitoring of liver glycerolipid partitioning in an animal nutrition biomodel fed with fish oil or a diet with an n-6/n-3 fatty acid ratio of 4:1 reported that blood lipids can be lowered by controlling the switch in the direction of the fatty acyl-CoA moiety flux to TAG and PL secretion [12]. However, research findings on in vivo monitoring of a direct relationship between egg intake and partitioning of hepatic glycerolipid, which is secreted from the liver into blood, are not yet available.

The present research was designed to obtain more information on the molecular mechanism of the biochemical metabolic pathway between egg intake and the partitioning of hepatic glycerolipid in reducing blood lipids using in vivo monitoring techniques from animal nutrition biomodels fitted with a jugular-vein cannula after smart-farm egg intake. The emerging research findings may suggest that the movement switch of fatty acyl-CoA moieties for regulating oxidation and esterification of hepatic glycerolipid address the direction of phospholipid action in the egg yolk after egg intake, thereby lowering blood lipids. This study dispels consumer misconceptions about cholesterol from eggs and provides molecular biology evidence that can promote egg consumption and lower blood lipids.

## 2. Materials and Methods

### 2.1. Experimental Design

The animal experiments were approved by the Institutional Animal Care and Use Committee (IACUC) of Kangwon National University, Republic of Korea (KW-200172; the approval date of the ethics committee, 1 March 2020). Male Sprague–Dawley rats (n = 24, body weight 180–190 g) were used as the lipoprotein-recipient animals for inserting the jugular-vein cannula, and old female rats (n = 3, average body weight 600 g) were used as donor animals for the preparation of the lipoprotein solution. Both these types of rats were purchased from Daehanbiolink Co. Ltd. (Chungcheongnam-do, Republic of Korea). The animals were orally administered the substances corresponding to each treatment group at a fixed time every morning for 40 days from the start of the experiment to the end of the experiment after a one-week environmental adaptation period by using a zonde. Four treatment groups (six animals per treatment, one animal per cage) were placed in a completely randomized block design: CON (control group, oral administration of 1.0 g of saline), T1 (oral administration of 1.0 g of pork belly fat), T2 (oral administration of 1.0 g of egg yolk), and T3 (oral administration of T1 and T2 alternately every week). T3 was designed to compare the partitioning effect on glycerolipid metabolism when pork belly fat and egg yolk were alternately consumed every week versus the results of eating egg yolk and pork belly fat alone. Egg yolk (smart-farm egg yolk) was used in here and was isolated from eggs that were produced by a digital livestock system [9]. The levels of oral administration of egg yolk and pork belly fat were determined to be plateaued when the level of blood lipids was no longer reduced when egg yolk and pork belly fat were orally administered by conducting a preliminary experiment (0, 0.5, 1.0, and 1.5 g oral administration). The pork belly fat used in this study was in the form of drippings obtained by roasting pork belly in a frying pan. The levels of cholesterol and SFAs were confirmed as 108 mg/100 g and 46.39 g/100 g in the pork belly fat, and 233 mg/100 g and 34.62 g/100 g in the egg yolk, respectively, as per the published analysis methods [1] (Table 1). 

### 2.2. Experimental Diet and Feeding Management

The animals were kept in separate cages, and tap water as well as purified diets (AIN-93, Flemington, NJ, USA) were provided ad libitum. A purified pellet diet adjusted to meet the nutritional requirements of rats, as prescribed by the American Institute of Nutrition 93 (AIN-93), was used as the basal diet. The basal diet consisted of corn sugar (39.80%), lactic casein (20.00%), granular sugar (10.00%), dextrin (13.20%), soy oil (7.00%), solka floc-40 (5.00%), AIN-93 mineral mix (3.50%), AIN-93 vitamin mix (1.00%), L-cystine (3.00%), and choline bitartrate (0.25%). The basal diet contained crude protein 20.00%, crude fat 7.00%, crude fiber 5.00%, crude ash 3.15% and AIN-93 vitamin–mineral mix. The animals were maintained at a temperature of 22 °C, relative humidity of 55%, and a 12 h light/dark cycle. During the experiment, the food and water intake, body weight, and other physical indicators (body condition, eyes, skin, hair, perineum) of the animals were monitored in the same manner as conducted by Park and Park (2016) in a previous study [1]. Each animal was weighed weekly, and each food and water intake was measured daily throughout the intervention trial period.

### 2.3. In Vivo Monitoring

The basic principle of in vivo monitoring is to determine the biochemical metabolic partitioning of newly synthesized hepatic glycerolipids secreted from the liver into the blood and tissues as a result of the injection of isotope-labeled lipoprotein solutions (ILPs), i.e., [^3^H]-cholesteryl oleyl ether (hereafter referred to as ^3^H) and cholesteryl-[^14^C]-oleate (hereafter referred to as ^14^C lipid), through a jugular-vein cannula. The oxidation (^14^CO_2_ emission) and esterification (TAG and PL synthesis from fatty acids) of hepatic glycerolipids were measured. Moreover, ^14^CO_2_ emission through the respiratory metabolic trials was calculated by collecting the metabolic amount after injection of the ILPs. Since VLDL-TAG, which was newly synthesized in the liver, was transferred to the blood within 30 min after injecting the ILPs, the oxidation and esterification rates of fatty acids could be measured accurately (Figure 1). To brief Figure 1, (a) a mixed population of VLDL- and chylomicron-remnants, rich in apoprotein E (apoE) but poor in apoC, is generated in a donor animal that is functionally hepatectomized and injected with heparin. The lipoproteins are separated by centrifugation and (b) the cholesterol ester (C-FA) pool in the hydrophobic core of the remnants is labelled by incubation in vitro with cholesteryl-[^14^C]-oleate (C-FA*) and ^3^H-cholesteryl oleoyl ether in the presence of the cholesterol ester transfer protein (CETP) in human plasma. (c) After re-isolation, the lipoproteins are injected into the experimental animals through jugular cannulae. (d) The cholesteryl ester is preferentially transferred into hepatocytes as the lipoprotein particles interact with hepatic lipoprotein lipase and lipoprotein receptors. (e) The cholesteryl ester is hydrolyzed by esterase to generate ^14^C-oleate, which labels the fatty acids of the liver cells. The acyl CoA partitioning between oxidation and esterification is studied by monitoring the exhaled ^14^CO_2_ (generated directly in the liver and indirectly after oxidation of ketone bodies in peripheral tissues) and the labelled triacylglycerol that is secreted or stored in the cytosol (TAGc). The labelled phospholipid is largely retained in cellular membranes. 

An in vivo cannulation technique was utilized in accordance with the European laboratory animal handling license (SCT-W94058) acquired by the author. The cannula was inserted after 40 days of oral administration of egg yolk; i.e., 5 days before ILP injection. The animals were anesthetized via an intraperitoneal injection of 0.15 mL mixture of ketamine (50 mg/mL, ketamine hydrochloride, Yuhan Chemical Inc., Republic of Korea) and rumpun (23 mg/mL, xylazine hydrochloride, Bayer, Republic of Korea) at a ratio of 3:1 per 100 g body weight. A sterilized polyethylene catheter (ID: 0.63 mm, OD: 1.19 mm; Silastic tubing, VWR–Dow Corning No 508-003, Midland, MI, USA) was inserted into the right internal jugular vein of each animal. In order to check the blood flow, cannula flushing was performed three times a day using 0.3 mL of citric saline. The blood flow was checked periodically, and a recovery period of 5 days was allowed.

Furthermore, 48 h before blood collection, 10% fructose solution was administered to the animals to accelerate new synthesis and the secretion of VLDL-TAG from the livers of donors. Then, 5 mL of blood was collected via the abdominal aorta after anesthetizing the donor animals, and it was used for lipoprotein isotope labeling. To accelerate the degradation of remnant lipoproteins (LDL), which constitute a sub-fraction of TAG-rich lipoproteins (VLDL-TAG), heparin was added to the blood obtained from the donor animals and incubated with the reaction reagent (1 µg/mL leupeptin 50 µL + 0.5% bovine serum albumin 0.25 mL + 3.2 mM EDTA 50 µL) for 30 min at 37 °C. The culture fraction was adjusted to a density of 1.019 g/mL with KBr (to obtain an intermediate-density fraction, but to exclude low-density lipoprotein) and centrifuged at 105,000× *g* for 18 h at 12 °C. The top 1 mL, containing the apoC-depleted lipoproteins, was collected and dialyzed against several changes of 0.15 M-NaCl containing 1.3 mM EDTA and 0.3 mm thimerosal, followed by dialysis against EDTA-free medium. Then, 2 mL of the IDL fraction was passed using size-exclusion chromatography (Sephadex G25 column) to remove KBr. IDL was passed through the above column, and 9 μL of [^3^H]-cholesteryl oleyl ether (1 mCi/mL, ARC, Inc. St. Louis, MO, USA) and 7.5 μL of cholesteryl-[^14^C]-oleate (50 µCi, Perkin Elmer, Waltham, MA, USA) were placed in one quick fit small tube and dried under nitrogen gas. Then, 1 mL of acetone was added and mixed slowly. After adding 1.5 mL of cholesterol ester transfer protein (CETP) (Sigma Aldrich, St. Louis, MO, USA), isotope-labeled LPS was prepared by bubbling and mixing for 15 min using nitrogen gas. CETP was added to incorporate apoC, which was not present in rat blood, into IDL. After re-isolution, the remnant were freed of extraneous label by size-exclusion chromatography (Sephadex G25) and dialyzed against saline. The prepared ILP (apoC-poor lipoproteins; chylomicron and VLDL remnants) was filtered through a 0.45 μm filter pore size immediately before use.

The amount of isotopically labeled ILP injected into the animal was adjusted to ^3^H 300,000 dpm (disintegrations per minute) and ^14^C lipid 300,000 dpm using a scintillation counter (Packard 1600TR, Hewlett Packard, Palo Alto, CA, USA). Then, 250 μL of ILP was injected through the jugular-vein cannula, followed by injection of 1.0 mL of 10% Triton WR 1339 (Sigma-Aldrich, St. Louis, MO, USA) dissolved in saline solution to accelerate the secretion of lipids from the liver into the blood. The animals were placed in a respiratory desiccator chamber (Pump by Masterflex model 7524-50, Cole-Parmer Instrument Co. Ltd. Vernon Hills, U.S.A) supplied with air (at a rate of 5 L/min per animal), and then the respiratory metabolism experiment was performed for 60 min. The ^14^CO_2_ was collected using 100 mL of an alkaline absorbent (directly prepared with ethanolamine and ethylene glycol monoethyl ether mixed solution, 1:2, *v*/*v*) for 60 min. The oxidation rate (^14^CO_2_ emission) was then calculated.

### 2.4. Sampling and Thin-Layer Chromatography

The animals were anesthetized via an intraperitoneal injection of pentobarbitone (60 mg/kg of body weight). Blood samples (3 mL) were obtained from the abdominal aorta while maintaining the animal’s normal body temperature using infrared light. The liver, adipose tissue, and hind leg muscles obtained from the animals were rapidly frozen in liquid N_2_ gas and stored at −80 °C until the biochemical analyses. The lipids were extracted using a chloroform/methanol 2:1 solution from each tissue and serum, and the lipid fractions (TAG and PL) were separated by thin-layer chromatography (TLC; Merck KGaA, Darmstadt, Germany).

### 2.5. Statistical Analyses

Statistical analyses of the obtained data were carried out using SPSS software version 20.0 (IBM Inc., Armonk, NY, USA). Considering egg yolk as the main factor, comparisons were performed between groups using general linear modeling. Statistically significant differences (*p* < 0.05) in the average values of the treatment groups were verified using one-way analysis of variance (ANOVA) and Duncan’s multiple range test. The results are expressed as the mean and pooled standard error of mean (PSE) of the animal nutrition biomodels. The number of animals used in each treatment group is indicated in the legends of all the tables in the paper (n = 6). 

## 3. Results

### 3.1. Incorporation into Liver and Tissue Accumulation of ^3^H and ^14^C Lipid

The incorporation of ^3^H remaining in the liver that was not secreted into the blood by hepatocyte metabolism after injecting the ILPs into the jugular vein cannula was found to range from 91.81% to 92.23% (Table 2). On the other hand, the incorporation of ^14^C lipid that was not metabolized in the liver (20.81–38.91%) after injecting the ILPs was significantly higher in the order of the T1, T3, CON, and T2 groups (*p* < 0.05). The tissue ^14^C lipid accumulation was statistically significantly different between each treatment group in the plasma, adipose tissue, and total muscle (*p* < 0.05). The tissue ^14^C lipid accumulation was significantly lower in the T1 group that received oral administration of pork belly fat than in the other groups, but it showed a higher tendency in the CON and T2 groups that received oral administration of egg yolk (*p* < 0.05).

### 3.2. Oxidation and Esterification of Hepatic Glycerolipid

The hepatic glycerolipid partitioning (esterification and oxidation of ^14^C lipid) after injecting the ILPs into the jugular-vein cannula was faster in the egg yolk group than in the pork belly fat group (Table 3). The level of total glycerolipids (% of cholesteryl-^14^C-oleate metabolized in the liver) was significantly higher in the order of T2, T3, and T1 = CON groups (*p* < 0.05). The levels of phospholipid (% of total glycerolipid labelled) secreted into the blood, % phospholipid/total glycerolipid, and % ^14^CO_2_/glycerolipid were significantly highest in the T2 group (*p* < 0.05), but the level of TAG (% of total glycerolipid labelled) secreted into the blood was highest in the T1 group (*p* < 0.05). 

### 3.3. Quantification to the Partitioning of Acyl Moieties Flux for Esterification of Hepatic Glycerolipid

Figure 2 provides an explanation based on the data presented in Table 2. It was observed that the influx of fatty acyl-CoA moieties in the direction of TAG synthesis from the injected ILPs after egg yolk intake was decreased, whereas the influx into PL synthesis was increased in vivo. In the biochemical metabolic pathway of fatty acyl-CoA moieties for the esterification (PL, TAG from ^14^C lipid) of hepatic glycerolipids, the flux of fatty acyl-CoA moieties in the direction of TAG synthesis showed a trend to be lower in the order of the T2, CON = T3, and T1 groups. However, the flux of fatty acyl-CoA moieties toward PL synthesis was found to be high.

## 4. Discussion

In a previous study using normal rats, Park and Park (2016) have reported that when designed under the same conditions and methods as this experiment, the variation trend of the food and water intake, body weight, serum metabolites, visceral index, and other physical indicators of the animals has significantly improved in the egg yolk group [1], which supports this result. Table 3 shows the incorporation of ^14^C lipid that was not metabolized in the liver after injecting the ILPs into the jugular-vein cannula. In the egg yolk group, the hepatic glycerolipid metabolism after injecting the ILPs was faster than that in the other groups. During previous in vivo monitoring, the report of faster glycerolipid metabolism in a second generation, animal nutrition biomodels, or obese animal models fed with dietary n-6/n-3 fatty acids at a 4:1 ratio showed a similar trend as this result [12]. The ^3^H in Table 2 was used to correct the metabolic rate of total glycerolipid synthesized and secreted in the liver from Table 3. Unlike cholesteryl-^14^C-oleate, ^3^H-cholesteryl oleoyl ether did not undergo the same metabolic pathway in the liver; thus, more than 91.81% persisted in the liver (Table 2). For this reason, incorporation of ^3^H in the liver is important for correcting the metabolic distribution of ^14^C lipid [12]. Cholesteryl-^14^C-oleate is metabolized into ^14^C-oleic acid and cholesterol in the liver; oleic acid is used to synthesize VLDL-TAG through oxidation and esterification, which then travels to the blood within about 30 min. Glycerolipid is newly produced in the liver after the injection of ^14^C lipid-labeled ILPs, and then when lipid metabolism progresses, the incorporation (liver residual rate) of ^3^H is much higher than that of ^14^C lipid, but the secretion of ^3^H into the blood is much lower or close to zero (Table 2). When ^3^H and ^14^C lipid are to be used together to assess the in vivo monitoring of the biochemical metabolic pathway of hepatic glycerolipid distribution, the amount of newly synthesized VLDL-TAG secreted from the liver into the blood can be accurately measured. Therefore, it can help to identify the molecular mechanism of hepatic glycerolipid partitioning [12,13,14].

Table 3 indicates the incorporation of cholesteryl-^14^C-oleate in the liver (% cholesterol ester not metabolized in the liver), i.e., hepatic glycerolipid partitioning corrected using the recovery values of ^3^H-cholesteryl oleoyl ether presented in Table 2. It presents the metabolized amounts in percentages based on the ^3^H recovery values to correct the differences in the measured values based on lipid metabolism. These results demonstrate that egg yolk (phospholipid) could increase PL secretion and ^14^CO_2_ emission rather than TAG secretion that moved into the blood. These results show that egg yolk can lower TAG synthesis in the liver and secretion into the blood and increase PL secretion and ^14^CO_2_ emission in the partitioning of newly synthesized glycerolipids in the liver. Moreover, ^14^CO_2_ emission (% ^14^CO_2_/glycerolipid) is more important than the oxidation of ^14^C-ketone bodies during in vivo monitoring of glycerolipid metabolism [12]. The ^14^C lipid accumulation in the muscle, as shown in Table 2, was the lowest in the egg yolk group, which was attributed to the increase in ^14^CO_2_ emission along with a decrease in secreted TAG (% of total glycerolipid) into the blood, shown in Table 3. Muscle is important for the clearance of circulating TAG for oxidation. Therefore, it can be concluded that egg intake can reduce lipid accumulation in tissues by inhibiting lipid absorption while promoting oxidation (Table 3) [15,16]. Blood lipids are regulated by the movement of newly synthesized glycerolipids from the liver into the blood. When the esterification rate of fatty acids (TAG synthesis) by the liver decreases, PLs are mostly maintained in the liver to maintain an appropriate cell membrane structure and conserve energy for PL synthesis [17,18]. Blood lipids are regulated by the movement of newly synthesized glycerolipids (TAG and PLs) from the liver into the blood. The activation of the partitioning of the biochemical metabolism of hepatic PLs lowers the blood TAG and cholesterol [19,20]. In the dose effect of egg, an intake of 1 g of egg yolk per day was associated with a plasma 2.29% accumulation of ^14^C lipid of the injected dose (Table 2), a 25.24% partitioning of ^14^C lipid of metabolized in liver, and 18.35% ^14^CO2 emission (Table 3). These findings suggest that consumption of eggs (1.0 g of egg yolk per day) as part of a healthy diet in animal nutritional biomodels may be beneficial for lipid metabolism, and that egg consumption may also be recommended in healthy human diets without concerns about lipids.

The highlights of the results are presented in Figure 2, which shows that lowering blood lipids via hepatic glycerolipid partitioning in vivo occurs by the action of phospholipids (lecithin and choline) present in eggs. It can be seen that egg yolk achieves a significant reduction in the proportion of label that is secreted as ^14^C-TAG from the liver compared with pork belly fat, with intermediate values for the other groups. It can be computed that for the egg yolk, the overall effect is achieved by the multiplicative decreases (compared with the pork belly fat) in the degree of partitioning of acyl-CoA, DAG and TAG, respectively, toward that portion of the acyl moiety that follows the pathway leading to TAG secretion. Therefore, the increased diversion of acyl-CoA toward oxidation and the increased intracellular retention (cytosol) of TAG at the expense of secretion make an approximately equal contribution, whereas the effect of the increased diversion of DAG toward phospholipid synthesis contributes about 32% of the overall effect (Figure 2). A previous study confirmed that egg intake lowered the blood lipids via modulating the biochemical metabolic mechanism through egg phospholipids [1]. Similarly, the flux of fatty acyl-CoA moieties at the three branch points (DAG, PL, and TAG secretion) of the metabolic pathway involved in glycerolipid synthesis is increased toward PL synthesis. The blood TAG and cholesterol levels were reduced in an obese animal biomodel fed with dietary n-6/n-3 in a 4:1 ratio [12]. The research findings confirmed that egg yolk (phospholipid) could also lower TAG and increase PLs in the biochemical metabolic pathway of hepatic glycerolipid partitioning. The decrease in TAG in the T2 group (oral administration of egg yolk) compared to the T1 group (oral administration of pork belly fat) was attributed to multiple metabolic pathways, which regulate the movement switch of fatty acyl-CoA moieties forward of DAG, PLs, and TAG. Therefore, it can be seen that the switch of fatty acyl-CoA moieties was actuated in the direction toward esterification (decreased TAG accumulation, and increased PL synthesis) in the liver after egg yolk intake (Figure 2) [16,21]. Synthesis of glycerolipids is the main route of fatty acid metabolism in the liver of animal nutrition biomodels. Tissues use very small amounts of the newly synthesized fatty acids for partial esterification of glycerols [22,23]. PLs are the major nutrients in the digestive tract after triacylglycerol. Egg yolk PLs are hydrolyzed by phospholipase in the small intestine, resulting in a lysophospholipid and a fatty acid part. Through the formation of micelles and passive diffusion, the lysophospholipids are directly absorbed by the small intestine epithelium. The lysophospholipid, particularly phosphatidylcholine (PC), play a key role in chylomicron synthesis and secretion [24,25,26,27]. During the biochemical metabolism of lymph chylomicrons, a proportion of chylomicron PLs are transferred to high-density lipoprotein (HDL) via phospholipid transfer protein [28,29]. The chylomicron PC after its transfer to HDL (HDL-PC) is hydrolyzed by hepatic lipase, metabolized as intact PC. In human and animals, HDL-PC is metabolized without rapid equilibration with LDL, VLDL, and erythrocyte membrane PC [30,31,32,33].

Hepatic glycerolipid partitioning in the liver between the formation of acylcarnitine for the oxidation of fatty acids and esterification for lipogenesis decides the oxidation rate of fatty acids. Thus, the partitioning of fatty acids in the liver between oxidation and esterification is important [16]. The biochemical metabolism of fatty acids, as hepatic glycerolipids, in humans and animals occurs via esterification and oxidation. Hepatic glycerolipids, which are newly synthesized in the liver and secreted into the blood, are synthesized from fatty acids, and they are used for cell membrane components and energy metabolism [12]. The report that the blood lipid level was lowered by the modulation of hepatic glycerolipid partitioning in n/6n3 at a 4:1 ratio when in vivo monitoring was performed after feeding diets with various ratios of n-6/n-3 fatty acids (71:1, 4:1, 15;1, 30:1) to animal nutrition biomodels showed a similar trend as this result [12]. A limitation of this study is that it did not investigate the brain–gut axis or gut–liver axis communication related to energy and glucose homeostasis. The gut provides essential information to the brain related to nutrients can be appropriately adjusted energy and glucose homeostasis. The brain–gut axis or gut–liver axis is regulated by various hormones secreted from the gut in response to nutrients, and these signals may enter the circulation and act directly on the brain, or the enteric nervous system can act as a relay from the brain to the gut [34]. Meanwhile, this study did not consider the influence of the brain–gut axis or gut–liver axis on animal metabolism, and additional research in this regard is expected to be necessary in the future.

## 5. Conclusions

In conclusion, the results of this study indicated that egg intake (phospholipids in the egg yolk) could modulated hepatic glycerolipid partitioning, thereby lowering blood TAG and increasing PL levels in vivo in animal nutrition biomodels. And also, egg intake could lower lipid accumulation in the liver, adipose tissue, and muscle, and increase CO_2_ emission by upregulating the fatty acid oxidation in the liver. The upregulated switch of the influx of fatty acyl-CoA moieties indicated that the partitioning of the newly synthesized glycerolipids in the liver was accelerated in the direction of PL synthesis and secretion by egg intake. These findings could provide a novel perspective on the molecular mechanism of hepatic glycerolipid partitioning through which eggs lower blood lipids. In the future, additional research is needed on the brain–gut axis or gut–liver axis communication related to hepatic glycerolipid partitioning.

## Figures and Tables

**Figure 1 foods-13-00722-f001:**
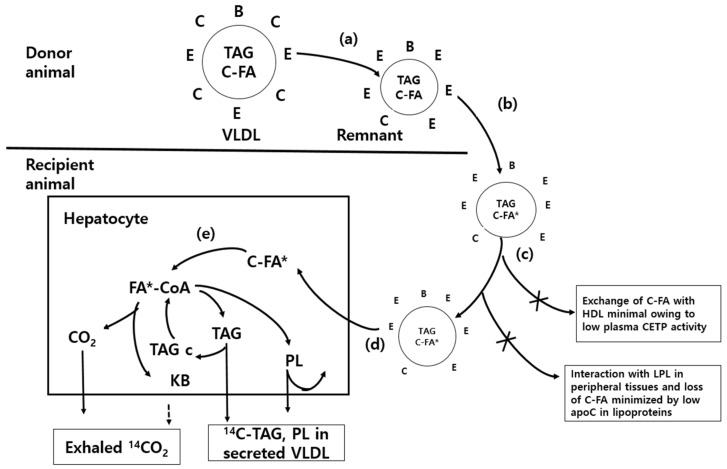
Basic principles of the in vivo monitoring technique for metabolic partitioning of hepatic glycerolipid in animal nutrition biomodels. * Indicates the metabolic pathway of cholesteryl-[^14^C]-oleate (TAG* C-FA), which is broken down from VLDL-TAG newly synthesized by cholesteryl-[^14^C]-oleate in the liver. B, C and E indicated apolipoproteins. (a) A mixed population of VLDL- and chylomicron-remnants. (b) Lipoprotein separation. (c) Lipoprotein injection into jugular-vein cannula. (d) Cholesteryl ester transferred into hepatocytes. (e) ^14^C-oleate generate cholesteryl ester via hydrolysis by esterase. HDL: high-density lipoprotein, CETP: cholesteryl ester transfer protein, LPL: lipoprotein lipase, TAG: triacylglycerol, PL: phospholipid, VLDL: very-low-density lipoprotein, KB: ketone body.

**Figure 2 foods-13-00722-f002:**
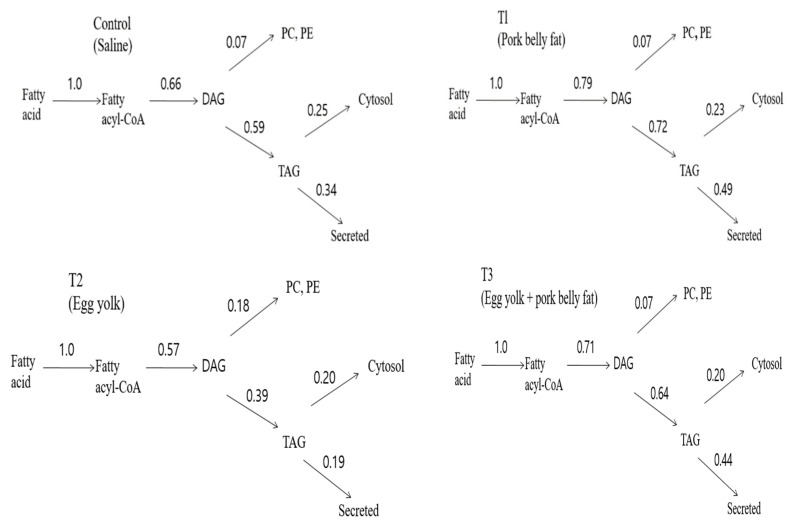
Partitioning of fatty acyl-CoA moieties fluxes for esterification of glycerolipids in liver in animal nutrition biomodels after oral administration of egg yolk. CON (control group, oral administration of 1.0 g of saline), T1 (oral administration of 1.0 g of pork belly fat), T2 (oral administration of 1.0 g of egg yolk), T3 (oral administration of T1 and T2 alternately every week). DAG: diacylglycerol, TAG: triacylglycerol, PC: phosphatidylcholine, PE: phosphatidyl ethanolamine.

**Table 1 foods-13-00722-t001:** Fatty acid profiles of the egg yolk and pork belly fat (% of total fatty acids).

	Egg Yolk	Pork Belly Fat
Myristic acid (C14:0)	0.53	2.18
Palmitic acid (C16:0)	26.05	30.15
Palmitoleic acid (C16:1 n-7)	3.33	2.37
Stearic acid (C18:0)	8.03	14.02
Oleic acid (C18:1 n-9)	44.97	36.26
cis-Vaccenic acid (C18:1 n-7)	0.05	0.05
Linoleic acid (C18:2 n-6)	16.49	13.85
Linolenic acid (C18:3 n-3)	0.26	0.05
Arachidic acid (C20:0)	-	0.02
Eicosenoic acid (C20:1 n-9)	0.09	0.71
Arachidonic acid (C20:4 n-6)	0.09	0.20
Adrenic acid (C22:4 n-6)	0.10	0.12
Lignoceric acid (C24:0)	0.01	0.02
Total	100	100
Saturated fatty acid (SFA)	34.62	46.39
Unsaturated fatty acid (UFA)	65.38	53.61
UFA/SFA	1.87	1.16
n-6/n-3	64.15	283

**Table 2 foods-13-00722-t002:** Incorporation and tissue accumulation of ^3^H and ^14^C lipid into the liver in animal nutrition biomodels fitted with a jugular-vein cannula after oral administration of smart-farm egg yolk ^1^.

	Incorporation of ^3^H Lipid (%) in Liver	Incorporation of ^14^C Lipid (%) in Liver	Accumulation of ^14^C Lipid in Plasma and Tissues (% of Injected Dose per Animal)		
			Plasma	Adipose Tissue	Total Muscle
CON	92.10	26.22 ^c^	4.55 ^a^	5.32 ^a^	5.81 ^a^
T1	92.04	38.91 ^a^	4.61 ^a^	5.16 ^a^	5.48 ^a^
T2	92.23	20.81 ^d^	2.29 ^c^	3.12 ^c^	3.02 ^c^
T3	91.81	28.59 ^b^	3.03 ^b^	4.01 ^b^	4.35 ^b^
PSE ^2^	1.8056	0.6025	0.0663	0.0825	0.0485
*p*-value	0.0250	0.0137	0.0279	0.0140	0.0348

^1^ CON (control group, oral administration of 1.0 g of saline), T1 (oral administration of 1.0 g of pork belly fat), T2 (oral administration of 1.0 g of egg yolk), T3 (oral administration of T1 and T2 alternately every week). ^2^ PSE: pooled standard error of mean values. ^a,b,c,d^ Values within the same column with different superscript are significantly different (n = 6, *p* < 0.05).

**Table 3 foods-13-00722-t003:** Partitioning of hepatic glycerolipids in animal nutrition biomodels fitted with a jugular-cannula vein after oral administration of egg yolk ^1^.

	Partitioning ^14^C-Lipid (%) of ^14^C-Lipid		Metabolized in Liver	
	Total Glycerolipid	Total TAG	Phospholipid	^14^CO_2_ Emission
CON	71.53 ^b^	58.79 ^a^	12.74 ^b^	11.73 ^b^
T1	57.72 ^d^	49.25 ^c^	8.47 ^d^	7.46 ^d^
T2	77.44 ^a^	52.19 ^b^	25.24 ^a^	18.35 ^a^
T3	68.86 ^c^	58.53 ^a^	10.33 ^c^	9.03 ^c^
PSE ^2^	2.3062	1.6583	0.4038	0.3401
*p*-value	0.0290	0.0311	0.0205	0.0078

^1^ CON (control group, oral administration of 1.0 g of saline), T1 (oral administration of 1.0 g of pork belly fat), T2 (oral administration of 1.0 g of egg yolk), T3 (oral administration of T1 and T2 alternately every week). ^2^ PSE: pooled standard error of mean values. ^a,b,c,d^ Values within the same column with different superscript are significantly different (n = 6, *p* < 0.05).

## Data Availability

The original contributions presented in the study are included in the article; further inquiries can be directed to the corresponding author.

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
