# Peer review of "In Vivo Monitoring of Glycerolipid Metabolism in Animal Nutrition Biomodel-Fed Smart-Farm Eggs"

_foods, 2024, doi:10.3390/foods13050722_

Round 1
Reviewer 1 Report
Comments and Suggestions for Authors
This study indicated that egg yolk can lower blood lipids by switch of the influx of fatty acyl-CoA moieties, which regulate the partitioning of newly synthesized glycerolipids in the liver. The research paper is quite meaningful, but some details need to be adjusted.
Line 4: important.
Line 168: Figure 1, Some words have a red underline, please remove the underline.
Line 210: Please unify whether the first letter of a word in the title is capitalized.
Line 374: The conclusion section should be more concise.
Line 394: The text in this section should be in a format that aligns both ends.
Line 461: Please unify connectors.
Comments on the Quality of English Language
This study indicated that egg yolk can lower blood lipids by switch of the influx of fatty acyl-CoA moieties, which regulate the partitioning of newly synthesized glycerolipids in the liver. The research paper is quite meaningful, but some details need to be adjusted.
Line 4: important.
Line 168: Figure 1, Some words have a red underline, please remove the underline.
Line 210: Please unify whether the first letter of a word in the title is capitalized.
Line 374: The conclusion section should be more concise.
Line 394: The text in this section should be in a format that aligns both ends.
Line 461: Please unify connectors.
Reviewer 2 Report
Comments and Suggestions for Authors
In this manuscript, the authors investigated the effect of egg yolk on glycerolipid metabolism as compared with pork belly fat. It is a very interesting paper. However the manuscript has some major flaws:
1. Is the observed effect only associated with smart farm-egg? It is not necessary to stress “smart farm-egg” in the title.
2. Line 25-27, what is the evidence for “egg lowers blood lipids”? Please list references. In the introduction, line 39-41, Ref 3 and 5 are wrongly cited, because both ref 3 and ref 5 did not support the idea that “egg intake could help reduce unhealthy blood lipids”.
3. Line 125, isotope-labeled lipoprotein solution was abbreviated as “LPS”. This is very confusing. In general, “LPS” refers to “lipopolysaccharide” in all literature.
4. In Figures 1 legend, line 172 “A-E indicated apolipoproteins” maybe replaced by “B, C, and E indicated Apo lipoproteins”. It is also necessary to briefly describe (a), (b), (c), (d), and (e).
5. In Conclusions Line 392, since the authors did not measure plasma lipid profiles, it is not appropriate to conclude “consuming eggs can lower blood lipid levels”.
Minor problems:
1. Line 32, 90, 94. “OOO” must be typos. Please correct.
2. In Materials and Methods 2.1, line 108-109, the level of cholesterol and SFAs in pork belly fat and eggs should be in the same unit, for example, mg/100g.
3. Ref 4, “Efeect” is spelled wrongly.
Reviewer 3 Report
Comments and Suggestions for Authors
This study evaluated the molecular mechanism of the biochemical metabolic pathway between egg intake and partitioning of hepatic glycerolipid in reducing blood lipids using in vivo monitoring techniques, which may provide evidence that dispels consumer misconceptions about cholesterol from egg. However, some clarification and improvements should be made before acceptance, especially the methods and results in this study.
1. Experimental Design: In addition to the ingredient and nutrient composition of the base diet, the authors should also provide the composition of egg yolk and pork belly fat used in the experiment, especially the type and content of fatty acids. For example, how many SFA and UFA are in the egg yolk and pork belly fat, respectively? The authors mentioned "smart farm-egg" many times in the paper. So, please explain the difference between the "smart farm-egg" and ordinary egg, and why the "smart farm-egg" is used.
2. I can’t get the design of T3 (oral administration of T1 and T2 alternately every week) group. Comparing to T1 and T2, it seems that T3 consumed half of the egg yolk and pork belly fat. What is the significance of this control?
3. The author should provide more data about the experimental animals. For example, what is the variation trend of the food and water intake, body weight, serum metabolites, visceral index, and other physical indicators of the animals in each group?
4. What does the author consider about the effects of the brain-gut axis or entero-liver axis on animal metabolism? Do the egg yolk and pork belly fat used in this study have a dose effect?
5. Although the authors tracked the patterns of fat metabolism, I think it is inadequate to speculate on the effects of egg consumption on the lipid metabolism which closely related to animal energy metabolism without the data about the food and water intake, body weight, visceral index, and other physical indicators of the animals.
Reviewer 4 Report
Comments and Suggestions for Authors
The manuscript is poorly written, and its content isn't easy to understand.
Since the Abstract presents a short overview of the manuscript, it should provide all relevant data, from background, aims, methodology, and results to conclusions. The first sentence is nonunderstandable. Is it necessary for consumers to be familiar with glycerolipid metabolism to consume eggs?
In this context, the Introduction started with glycerolipids' metabolic pathways instead of a logical introduction to potential future readers of the research aim and its relevance.
Did other studies investigate the same topic? What is the novelty of this study? What are smart farm eggs?
The experimental design must be improved. Consider preparing a graphical presentation for experimental groups, receiving treatments, and the duration of the experiments.
How were the diet and weight of animals monitored during the study?
What are the limitations of this research?
The conclusion should be more concise and should address further research directions.
Comments on the Quality of English Language
The manuscript needs extensive professional English editing.
Round 2
Reviewer 4 Report
Comments and Suggestions for Authors
The current version of the manuscript is improved compared to the initially submitted version. However, there are yet several points that should be addressed.
Change the explanation within the bracket with a short paragraph on smart-farm eggs.
In addition to citing the previous study within 2.2, add monitoring data relevant to this study (food and water intake, body weight, and other physical indicators of the animals, at least as supplementary material).
A conclusion should summarize the obtained study data, avoiding speculation and generalization.
Comments on the Quality of English Language
Some minor changes are needed.
